# Isolation, Antimicrobial Susceptibility, and Genotypes of Three *Pasteurellaeae* Species Prevalent on Pig Farms in China Between 2021 and 2023

**DOI:** 10.3390/microorganisms13040938

**Published:** 2025-04-18

**Authors:** Fangxin Li, Xin Zong, Guosheng Chen, Yu Zhang, Qi Cao, Lu Li, Huanchun Chen, Zhong Peng, Chen Tan

**Affiliations:** 1National Key Laboratory of Agricultural Microbiology, College of Veterinary Medicine, Huazhong Agricultural University, Wuhan 430070, China; lifangxin@webmail.hzau.edu.cn (F.L.); zongx@webmail.hzau.edu.cn (X.Z.); chenguosheng@webmail.hzau.edu.cn (G.C.); zhangyu@webmail.hzau.edu.cn (Y.Z.); cq20100804@163.com (Q.C.); lilu@mail.hzau.edu.cn (L.L.); chenhch@mail.hzau.edu.cn (H.C.); 2Hubei Hongshan Laboratory, Wuhan 430070, China; 3Frontiers Science Center for Animal Breeding and Sustainable Production, The Cooperative Innovation Center for Sustainable Pig Production, Wuhan 430070, China

**Keywords:** *Pasteurella multocida*, *Glaesserella parasuis*, *Actinobacillus pleuropneumoniae*, serotypes, antimicrobial susceptibility, whole-genome sequencing, antimicrobial resistance genes, virulence factor genes, pigs, China

## Abstract

In this study, we demonstrated the serotypes of three *Pasteurellaeae* species including *Pasteurella multocida* (PM), *Glaesserella parasuis* (GPS), and *Actinobacillus pleuropneumoniae* (APP), which are prevalent on Chinese pig farms. We also investigated the resistance profiles of these three bacterial species against different antimicrobial agents that are used for treating infections caused by them in pigs. We identified several virulence factor genes that were highly frequently presented and were conserved among these three *Pasteurellaeae* species. This study provides valuable insights into the epidemiology of these bacterial pathogens, which have significant implications for swine health and antibiotic stewardship.

## 1. Introduction

The family *Pasteurellaceae* comprises various pathogens that present significant threats to animal and/or public health and safety. Among them, *Pasteurella multocida*, *Glaesserella parasuis* (formerly *Haemophilus parasuis*), and *Actinobacillus pleuropneumoniae* are common pathogens in the global pig farming industry, causing respiratory diseases in pigs of different ages and leading to high morbidity, mortality, and economic losses [1]. Of these three *Pasteurellaceae* species, *P. multocida* is a Gram-negative pathogen responsible for swine atrophic rhinitis and pneumonic pasteurellosis [2,3]. *P. multocida* strains can be classified into five capsular serogroups (A, B, D, E, and F) [4]. Current reports indicate that types A and D are the predominant *P. multocida* serogroups among pigs, with sporadic cases of serogroup F strains reported [5,6].

*G. parasuis* is also a Gram-negative bacillus that depends on nicotinamide adenine dinucleotide (NAD) and causes Glässer’s disease in young pigs, presenting a total of fifteen serotypes (serotypes 1–15) [7]. Among them, serotypes 1, 5, 10, and 12 are highly pathogenic, serotypes 2, 4, 8, and 15 are moderately pathogenic, and serotypes 3, 6, 7, 9, and 11 are non-pathogenic, commonly found in healthy swine upper respiratory tracts [8]. Currently, serotypes 1, 5, 7, and 10 are the major infectious serotypes worldwide [9]. *A. pleuropneumoniae* can be divided into fifteen serotypes [10]. Among them, serotypes 1, 7, and 9 are the dominant serotypes in China [1]. The virulence of different serotypes primarily depends on the expression levels of RTX cytotoxins ApxI, ApxII, ApxIII, and ApxIV [11]. In addition to the toxins, all three of these *Pasteurellaceae* pathogens also encode multiple virulence factors, contributing to their pathogenesis and fitness. These include adhesion factors such as the type 1 fimbriae (FimB, FimE, FimA, FimI, FimC, FimD, FimF, FimG, and FimH), flagella (FlgJ, FlgH, FlgG, FlgE, FlgD, and FlgM), pili (YkgK/ecpR, YagZ/EcpA, YagY/EcpB, YagX/EcpC, YagW/EcpD, YagV/ecpE, EbpA, EbpB, and EbpC), and non-fimbrial adhesin (EfaA), proteins involved in iron uptake, transportation, and metabolism (FepA, FepB, FepC, FepD, FepG, IucA, IucB, IucC, IucD, IroB, IroC, IroD, IroE, and IroN), proteins contributing to serratiochelin and analog production (EntA, EntB, EntC, EntD, EntE, and EntF), the type II secretion system (T2SS; GspB, GspC, GspD, GspE, GspF, GspG, GspH, GspI, GspJ, GspK, GspL, and GspM), and outer membrane protein (OmpA) [12,13,14,15].

Currently, antimicrobial therapy plays a crucial role in the prevention and control of the aforementioned three swine *Pasteurellaceae* pathogens. Sulfonamides, lincomycin, and penicillins are commonly used for preventive treatments against swine pasteurellosis, while tetracyclines are recommended for treating infections caused by *A. pleuropneumoniae* [3,10]. For swine *G. parasuis* infections, florfenicols and rifampicin are recommended options [9]. However, recent studies have indicated that strains isolated from pig farms in recent years exhibit reduced sensitivity to the aforementioned drugs [1,16]. For instance, it was found that clinically isolated strains in Denmark displayed a 7.6% resistance to tetracyclines [17], while studies in Hungary revealed that clinically isolated strains exhibited a 55.6% resistance to tetracyclines and a resistance ranging from 5% to 52.5% to enrofloxacin [18]. Reports have indicated that between 2008 and 2010, *G. parasuis* in southern China showed resistance rates of 58%, 45.5%, 41.1%, and 29.4% to trimethoprim/sulfamethoxazole, enrofloxacin, tetracyclines, and penicillins, respectively [19]. A number of antimicrobial resistance genes (ARGs) have been reported to be responsible for the acquisition of these resistant phenotypes, including *tet*(A), *tet*(B), *tet*(C), *tet*(G), *tet*(H), *tet*(L), *tet*(M), and *tet*(O) for tetracycline resistance, *bla*_CMY-2_, *bla*_OXA-2_, *bla*_PSE-2_, *bla*_ROB-1_, and *bla*_TEM-1_ for penicillin resistance, *strA*, *strB, aadA1*, *aadA14*, and *aadA25* for streptomycin resistance, *erm*(A), *erm*(C), *erm*(T), and *erm*(42) for macrolide resistance, and *qnrA1*, *qnrB6*, and *aac(6′)-Ib-cr* for quinolone resistance [20].

China is the largest country in swine breeding and pork consumption globally. *P. multocida*, *G. parasuis*, and *A. pleuropneumoniae* stand as prevalent pathogens on Chinese pig farms, showing top five isolation rates in China annually [1]. Consequently, it is crucial that we comprehend the epidemic distribution characteristics of these three pathogenic bacteria within Chinese pig farming. However, further investigation is needed into the antimicrobial resistance profiles, as well as the serotypes and genotypes, of these three important swine *Pasteurellaceae* pathogens prevalent in China, as the epidemiological and resistant data for them are still limited. Therefore, this study conducted isolation identification, antimicrobial susceptibility testing, and whole-genome sequencing on the aforementioned three pathogens from pig farms in twelve provinces in China over the past three years. We aimed to gain insights into the current prevalence, dominant serotypes/genotypes, and antimicrobial resistance profiles of these three swine *Pasteurellaceae* pathogens.

## 2. Materials and Methods

### 2.1. Sample Collection and Bacterial Isolation

Between January 2021 and December 2023, a total of 4190 lung tissues from pigs that had succumbed to respiratory symptoms were submitted by pig farms from 12 provinces in China to the Veterinary Diagnostic Laboratory of Huazhong Agricultural University (HZAUVDL) in Wuhan, China, for examination for *P. multocida*, *G. parasuis*, and *A. pleuropneumoniae*. Bacterial strains were isolated from the samples under sterile conditions using an inoculation loop and cultured on tryptic soy agar (TSA; BD, Franklin Lakes, NJ, USA) supplemented with 10 µg/mL of Nicotinamide adenine dinucleotide (NAD; Sigma, St. Louis, MO, USA) and 5% newborn bovine serum (NBS; Boster Bio., Pleasanton, CA, USA) at 37 °C for 24 h. A single bacterial colony was selected from each sample for purification, followed by identification and capsular serotyping using specific gene PCR amplification methods [1,6]. The isolated and identified bacterial strains were preserved in 600 µL of tryptic soy broth (TSB; BD, Franklin Lakes, NJ, USA) containing 10 µg/mL of NAD (Sigma, St. Louis, MO, USA) and 5% NBS (Boster Bio., Pleasanton, CA, USA), mixed with 400 µL of sterile glycerol, and stored at −20 °C for future use.

### 2.2. Antimicrobial Susceptibility Testing

The protocol for the broth microdilution method, as recommended by the United States Clinical and Laboratory Standards Institute (CLSI document VET01S), was utilized to determine the minimal inhibitory concentrations (MICs) of the tested antimicrobial agents against different *P. multocida* and/or *A. pleuropneumoniae* strains [21]. As there is no CLSI method available for broth microdilution susceptibility testing in *G. parasuis*, previously recommended methods were followed [22,23]. The antimicrobial agents examined in this study included sulfamethoxazole-trimethoprim (sulfonamides), ceftiofur (cephalosporins), ampicillin (penicillin), tetracycline (tetracyclines), tilmicosin (macrolides), enrofloxacin (fluoroquinolones), and gentamicin (aminoglycosides). All agents were procured commercially from MedChemExpress (Monmouth Junction, NJ, USA), and their solutions for antimicrobial susceptibility testing (AST) were prepared following CLSI document VET01S guidelines (for *P. multocida* and *A. pleuropneumoniae*) or the referenced articles (for *G. parasuis*) [22]. The AST results for *P. multocida* and/or *A. pleuropneumoniae* were interpreted using the CLSI breakpoints [21]. In cases where CLSI breakpoints were unavailable, the MIC data were not categorized as susceptible, intermediate, or resistant. Each antimicrobial agent underwent testing in triplicate, and the entire experiment was conducted independently three times. *Staphylococcus aureus* ATCC^®^ 29213 and/or *Escherichia coli* ATCC 25922 were utilized as quality control strains. Multidrug-resistant (MDR) strains are defined as bacteria resistant to more than three classes of antibiotics [24].

### 2.3. Illumina Sequencing and Bioinformatical Analysis

Bacterial genomic DNA was extracted using the Cetyltrimethyl Ammonium Bromide (CTAB) method and assessed for DNA concentrations, quality, and integrity using a Qubit Fluorometer (Invitrogen, Waltham, MA, USA) and a NanoDrop Spectrophotometer (Thermo Scientific, Waltham, MA, USA). Subsequently, DNA libraries were prepared with the NEBNext^®^Ultra™ II DNA Library Prep Kit (New England Biolabs, Ipswich, MA, USA) and sequenced on an Illumina Miseq™ platform (Illumina, San Diego, CA, USA). Raw reads underwent processing with Adapter Removal [25] to eliminate adapter contaminations and the SOAPec program (version 2.03, https://anaconda.org/bioconda/soapec, accessed on 12 November 2024) to filter out low-quality data. Clean reads were then de novo assembled using SOAPdenovo2 [26], and the assembled contigs were used for further bioinformatic analysis.

Antimicrobial resistance genes (ARGs) were identified using the Comprehensive Antibiotic Resistance Database (CARD) [27], while virulence factor genes were identified with the Virulence Factor Database (VFDB) [28]. Whole-genome single-nucleotide polymorphisms (WG-SNPs) were determined through genome-wide pairwise comparison using MAFFT (version 7.222) [29]. A phylogenetic tree was constructed using Parsnp software (version 1.2) [30]. Sequence data were deposited in NCBI under BioProjects PRJNA1082799 (*P. multocida*), PRJNA1080278 (*G. parasuis*), and PRJNA1079884 (*A. pleuropneumoniae*).

### 2.4. Statistical Analyses

Statistical analyses were performed to compare the resistant rates of a specific antimicrobial agent between two bacterial species using a Chi-squared test. *p*  <  0.05 was considered to be significant.

## 3. Results

### 3.1. Bacterial Isolation and Distribution of Serotypes/Genotypes

A total of 151 strains from the three *Pasteurellaceae* species were isolated from the 4190 pig samples delivered. The overall isolation rate was 3.60% (151/4190). Among these strains, 64 were identified as *P. multocida* (1.53%, 64/4190), 48 were *G. parasuis* (1.15%, 48/4190), and 39 were *A. pleuropneumoniae* (0.93%, 39/4190). Detailed information regarding sample collection and bacterial isolation is provided in Table 1.

Bacterial typing revealed two known capsular types for the 64 *P. multocida* strains, with type D being predominant (50.0%, 32/64), followed by type A (43.75%, 28/64). Notably, four *P. multocida* strains (6.25%, 4/64) were untypable (Figure 1). Among the 48 *G. parasuis* strains, serotype 5/12 was most common (47.92%, 23/48), followed by serotype 4 (25.0%, 12/28), capsular serotype 7 (20.83%, 10/48), and capsular serotype 2 (6.25%, 3/48) (see Figure 1). Three capsular serotypes were identified for the 39 *A. pleuropneumoniae* strains, including serotype 7 (35.90%, 14/39), serotype 1 (30.77%, 12/39), and serotype 15 (30.33%, 13/39) (see Figure 1).

### 3.2. Antimicrobial-Resistant Phenotypes

The distribution of MIC values obtained for the 151 strains of the three *Pasteurellaceae* species is detailed in Table 2. The results of the antimicrobial susceptibility testing indicated that among *P. multocida* strains, the proportions resistant to ampicillin, tilmicosin, tetracycline, and enrofloxacin were 93.75% (60/64), 64.06% (41/64), 43.75% (28/64), and 34.38% (22/64), respectively (Table 2). The MIC_90_ values for sulfamethoxazole-trimethoprim, ceftriaxone, and gentamicin were 512 µg/mL, 32 µg/mL, and 32 µg/mL, respectively (Table 2). Regarding *A. pleuropneumoniae* strains, the rates of resistance against ampicillin, tilmicosin, tetracycline, and enrofloxacin were 71.79% (28/39), 58.97% (23/39), 61.54% (24/39), and 10.26% (4/39), respectively (Table 2). The MIC_90_ values for sulfamethoxazole-trimethoprim, ceftriaxone, and gentamicin were 512 μg/mL, 32 μg/mL, and 32 μg/mL, respectively (Table 2). However, there was no significant difference in the resistant rates of the above-mentioned agents between *P. multocida* and *A. pleuropneumoniae* (*p* > 0.05).

Due to the absence of CLSI breakpoints for *G. parasuis*, the isolates could not be categorized as susceptible, intermediate, or resistant, and, consequently, the proportions of resistant isolates could not be determined. Among the antimicrobial agents tested, ampicillin, tetracycline, and enrofloxacin exhibited the lowest MIC_90_ values (8 µg/mL), while sulfamethoxazole-trimethoprim had the highest MIC_90_ value (µg/mL) (Table 2). Notably, the MIC_90_ values for the various tested antimicrobial agents against *P. multocida*, *G. parasuis*, and *A. pleuropneumoniae* strains isolated in this study were uniform (Table 2). However, ceftriaxone displayed a higher MIC_50_ value for *P. multocida* (32 µg/mL) compared to *G. parasuis* (32 µg/mL) and *A. pleuropneumoniae* (2 µg/mL) (Table 2). Moreover, tetracycline demonstrated a lower MIC_50_ value for *P. multocida* (1 µg/mL) than for *G. parasuis* (8 µg/mL) and *A. pleuropneumoniae* (8 µg/mL). Gentamicin exhibited a higher MIC_50_ value for *G. parasuis* (32 µg/mL) compared to *P. multocida* (4 µg/mL) and *A. pleuropneumoniae* (4 µg/mL), whereas tilmicosin displayed a lower MIC_50_ value for *G. parasuis* (16 µg/mL) than for *P. multocida* (128 µg/mL) and *A. pleuropneumoniae* (128 µg/mL). Enrofloxacin showed a lower MIC_50_ value for *A. pleuropneumoniae* (0.25 µg/mL) than for *P. multocida* (0.5 µg/mL) and *G. parasuis* (0.5 µg/mL) (Table 2).

### 3.3. Distribution of Antimicrobial Resistance Genes and Their Associations with the Resistant Phenotypes

The examination of ARGs identified 18 genes that confer resistance to various classes of antibiotics, including sulfonamides (*sul2*, *sul3*, and *dfrA12*), cephalosporins (*bla*_CTX-M_ and *bla*_TEM_), penicillin (*lnu*(F) and *bla*_OXA_), tetracycline (*tet*(A), *tet*(L), and *tet*(M)), macrolides (*mph*(A), *mef*(B), and *erm*(42)), quinolones (*qnrS1* and *oqxB*), and aminoglycosides (*aadA22*, *aph(3′)-Ia*, and *strA*) in the 151 strains of the three *Pasteurellaceae* species (Figure 2). Among them, *tet*(L) (100%, 151/151), *tet*(M) (100%, 151/151), *tet*(A) (95%, 144/151), *bla*_TEM_ (93%, 141/151), *sul2* (93%, 140/151), *aph(3′)-Ia* (85%, 129/151), *dfrA12* (76%, 115/151), *qnrS1* (74%, 112/151), *strA* (71%, 107/151), *sul3* (71%, 107/151), and *mef*(B) (70%, 105/151) exhibited a high frequency of identification, while *oqxB* (17%, 26/151) and *bla*_OXA_ (6%, 9/151) showed a low frequency of identification (Figure 3A).

All *P. multocida*, *G. parasuis*, and *A. pleuropneumoniae* strains showed high proportions (≥60%) of carrying several ARGs, including *tet*(L), *tet*(M), tet(A), *bla*_TEM_, *dfrA12*, *sul2*, *strA*, and *aph(3′)-Ia* (Figure 3B). However, the examination of several ARGs in *G. parasuis* was remarkably lower than those in *P. multocida* and/or *A. pleuropneumoniae* (Figure 3B). These included *aadA22* (GPS vs. PM/APP: 6% vs. 70%/64%), *qnrS1* (GPS vs. PM/APP: 38% vs. 95%/85%), *mph*(A) (GPS vs. PM/APP: 15% vs. 91%/82%), *mef*(B) (GPS vs. PM/APP: 19% vs. 98%/85%), *erm*(42) (GPS vs. PM/APP: 10% vs. 64%/69%), *lnu*(F) (GPS vs. PM/APP: 6% vs. 63%/56%), sul3 (GPS vs. PM/APP: 23% vs. 100%/82%), and *bla*_CTX-M_ (GPS vs. PM/APP: 48% vs. 75%/67%). The presence of *oqxB* and *bla*_OXA_ in these three *Pasteurellaceae* species was low. Notably, a weak association between the presence of ARGs and the resistant phenotypes was observed in this study. For example, although 100% of the *P. multocida* (PM) and *A. pleuropneumoniae* (APP) carried tetracycline-resistant genes *tet*(L) and *tet*(M), less than 65% of them demonstrated phenotypes of tetracycline resistance (PM: 43.75%, APP: 61.54%) (Figure 3B, Table 2). Similarly, over 80% of the *P. multocida* and *A. pleuropneumoniae* carried macrolide-resistant genes *mph*(A) and *mef*(B); only approximately 60% of the *P. multocida* and *A. pleuropneumoniae* were resistant to tilmicosin (PM: 64.06%, APP: 58.97%) (Figure 3B, Table 2).

### 3.4. Distribution of Genes Associated with Bacterial Fitness and Virulence

We proceeded to analyze the distribution of 58 genes associated with bacterial fitness and virulence across the 151 strains of the three *Pasteurellaceae* species (Figure 4). These 58 genes encode bacterial type 1 fimbriae (*fimB*, *fimE*, *fimA*, *fimI*, *fimC*, *fimD*, *fimF*, *fimG*, and *fimH*), proteins involved in iron uptake, transportation, and metabolism (*fepA*, *fepB*, *fepC*, *fepD*, *fepG*, *iucA*, *iucB*, *iucC*, *iucD*, *iroB*, *iroC*, *iroD*, *iroE*, and *iroN*), proteins contributing to serratiochelin and analog production (*entA*, *entB*, *entC*, *entD*, *entE*, and *entF*), the type II secretion system (T2SS; *gspB*, *gspC*, *gspD*, *gspE*, *gspF*, *gspG*, *gspH*, *gspI*, *gspJ*, *gspK*, *gspL*, and *gspM*), flagella (*flgJ*, *flgH*, *flgG*, *flgE*, *flgD*, and *flgM*), pili (*ykgK*/*ecpR*, *yagZ*/*ecpA*, *yagY*/*ecpB*, *yagX*/*ecpC*, *yagW*/*ecpD*, *yagV*/*ecpE*, *ebpA*, *ebpB*, and *ebpC*), non-fimbrial adhesin (*efaA*), and outer membrane protein (*ompA*). The results indicate that several of these genes, including *fimB*, *fimA*, *fimD*, *fimF*, and *fepG*, were found in all *P. multocida*, *G. parasuis*, and *A. pleuropneumoniae* strains examined in this study (Figure 4). Overall, genes related to type 1 fimbriae biosynthesis were highly prevalent among the three *Pasteurellaceae* species investigated. However, certain genes exhibited species-specific preferences, even if they belonged to the same category. For instance, *iroBCDEN* genes were identified in over 85% of the *G. parasui* strains but were present in less than 35% of the *P. multocida* strains and fewer than 20% of the *A. pleuropneumoniae* strains (Figure 4).

## 4. Discussion

Respiratory infections pose significant challenges in the Chinese pig industry, with *P. multocida*, *G. parasuis*, and *A. pleuropneumoniae* identified as the primary causative bacteria behind swine respiratory illnesses [31,32]. In this study, we conducted bacterial isolation of three *Pasteurellaceae* species—*P. multocida*, *G. parasuis*, and *A. pleuropneumoniae*—from the lungs of pigs afflicted with respiratory disorders across various Chinese provinces. These three *Pasteurellaceae* species, recognized as the primary agents behind swine respiratory ailments, rank among the top five frequently isolated bacteria on pig farms in China [1,31]. Notably, the overall isolation rates of these *Pasteurellaceae* species in our study are comparatively lower than those reported in other research conducted in China [1,32]. Several factors may contribute to these lower isolation rates, with sample freshness emerging as a significant consideration. Although all samples were promptly processed upon arrival, prolonged transportation distances could impact sample freshness and the successful isolation of the three bacterial species. Moreover, it is plausible that *P. multocida*, *G. parasuis*, and/or *A. pleuropneumoniae* might not be solely responsible for the symptoms observed in the diseased pigs whose lung samples were analyzed. This is particularly pertinent given that numerous other pathogens such as *Streptococcus suis*, *Mycoplasma pneumoniae*, porcine reproductive and respiratory syndrome virus (PRRSV), and porcine circovirus (PCV) are also capable of causing swine respiratory infections and are prevalent on Chinese pig farms [32].

Our investigation revealed that *P. multocida* serogroups A and D, *G. parasuis* serotypes 5/12 and 4, and *A. pleuropneumoniae* serotypes 7 and 1 were predominantly prevalent on pig farms in China, findings that align closely with those of other studies conducted in China [6,32]. Notably, these serotypes are also documented as commonly occurring types on pig farms in various other Asian countries [33,34,35]. It is important to also acknowledge the identification of several other serotypes. For instance, while *G. parasuis* serotype 7 may not be as frequently identified as serotypes 5/12 and 4 on farms, multiple studies have emphasized serovar 7 as a significant disease-associated serotype of *G. parasuis* [35,36,37]. Moreover, a small proportion of *P. multocida* strains were untypable, indicating capsular serotypes that do not fall within categories A, B, D, E, or F [38]. The presence of untypable capsular serotypes in swine *P. multocida* has been reported in studies conducted in China and other countries [31,38,39]. The existence of untypable capsules in *P. multocida* could be attributed to either a lack of available tests [31] or could potentially signify the emergence of novel capsular types.

Our antimicrobial susceptibility testing results revealed concerning findings regarding the resistance profiles of *P. multocida* and *A. pleuropneumoniae* strains. For instance, ampicillin and enrofloxacin are commonly employed in treating pasteurellosis in pigs [3]. However, a significant proportion (93.75%) of the *P. multocida* strains tested in this study exhibited resistance to ampicillin, while 34.38% showed resistance to enrofloxacin. Notably, the resistance rates of *P. multocida* strains against these two antimicrobial agents, as well as tilmicosin, were notably higher compared to those in a recent study [16], suggesting that *P. multocida* strains prevalent on different farms may display varying resistance patterns. While several studies have indicated that a substantial number of *A. pleuropneumoniae* clinical strains from Europe and North America are susceptible to ampicillin and enrofloxacin [40,41], our data in this study, along with findings from other studies, highlighted significant proportions of *A. pleuropneumoniae* clinical strains from pigs in Taiwan and Australia that were resistant to ampicillin and other β-lactams [20,42]. These results hint at the existence of complex resistance profiles among *A. pleuropneumoniae* strains. Furthermore, the resistance rates of *A. pleuropneumoniae* to other agents such as tilmicosin and tetracycline were comparable to those reported in other studies [41]. Although the resistant rates of *G. parasuis* against these tested agents could not be determined due to the lack of available breakpoints, our broth microdilution assays revealed that many agents demonstrated high MIC_90_ values against a high proportion of the *G. parasuis* isolates, which is consistent with previous studies from different countries [20,23], and these results indicate worrisome resistance conditions. As for the clinical management of infections caused by bacteria including the three *Pasteurellaceae* species investigated in this study, antimicrobial stewardship, bio-security improvement, and vaccination represent practical strategies to combat the increasing threat of bacterial antimicrobial resistance [43]. In addition, continuously monitoring the resistant phenotypes of clinical isolates is also important.

Our analysis did not reveal a significant correlation between the presence of ARGs and resistant phenotypes in the three *Pasteurellaceae* species. A previously published article also noted that ARGs in porcine *P. multocida* were not linked to its antimicrobial susceptibility pattern [44]. Similarly, in another study, the susceptibility to antimicrobials in *A. pleuropneumoniae* did not strongly align with genotypic patterns [42]. This suggests that the development of antimicrobial resistance in *Pasteurellaceae* species may involve mechanisms beyond ARGs [45]. In addition to ARGs, we also analyzed the distribution of VFGs among the three *Pasteurellaceae* species. Many genes encoding proteins involved in bacterial adherence (such as type 1 fimbriae, flagella, pili, non-fimbrial adhesin, or OmpA) were found to be widely carried by these species. Previous genome-based studies have demonstrated similarities in these proteins across the three *Pasteurellaceae* species [12,13,14,15]. Consequently, some of these proteins could serve as potential immunogenic candidates to combat infections caused by *P. multocida*, *G. parasuis*, and/or *A. pleuropneumoniae*, as bacterial adherence-associated proteins are considered promising targets for intervention strategies [46,47].

We should acknowledge that our study has several limitations. The samples used for bacterial isolation were delivered by farm owners, which may limit our collection of bacterial strains from more regions. In addition, the period is also relatively short, which may restrict our further understanding of the comprehensive change dynamics behind the antimicrobial resistance trends in these three *Pasteurellaceae* species. We intend to continuously monitor the antimicrobial resistance profiles of different bacterial species prevalent on Chinese pig farms, and we hope to provide a comprehensive picture to reflect the bacterial antimicrobial resistance in swine in future.

## 5. Conclusions

In conclusion, our study reported on the isolation, prevalent serotypes, antimicrobial susceptibility, and presence of ARGs and VFGs in three *Pasteurellaceae* species—*P. multocida*, *G. parasuis*, and *A. pleuropneumoniae*—from pigs in China. While these species exhibited diverse serotypes, certain types were detected at high frequencies. Our antimicrobial susceptibility testing highlighted significant levels of antimicrobial resistance in these *Pasteurellaceae* species, with a notable proportion of strains showing resistance to therapeutic agents. Although our whole-genome sequencing analysis identified various ARGs, a strong correlation between phenotypic and genotypic susceptibility to antimicrobials was not observed. The distribution of VFGs, particularly those involved in bacterial adherence, was widespread among the three species, suggesting potential targets for intervention strategies.

## Figures and Tables

**Figure 1 microorganisms-13-00938-f001:**
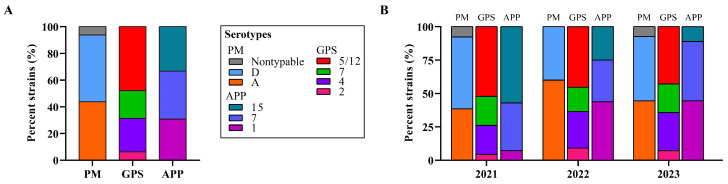
Distribution of bacterial serotypes of the three *Pasteurellaeae* species. (**A**) A column chart showing the distribution of serotypes of the three *Pasteurellaeae* species determined in this study. (**B**) A column chart showing the distribution of serotypes of the three *Pasteurellaeae* species in different years. PM, *Pasteurella multocida*; GPS, *Glaesserella parasuis*; APP, *Actinobacillus pleuropneumoniae*.

**Figure 2 microorganisms-13-00938-f002:**
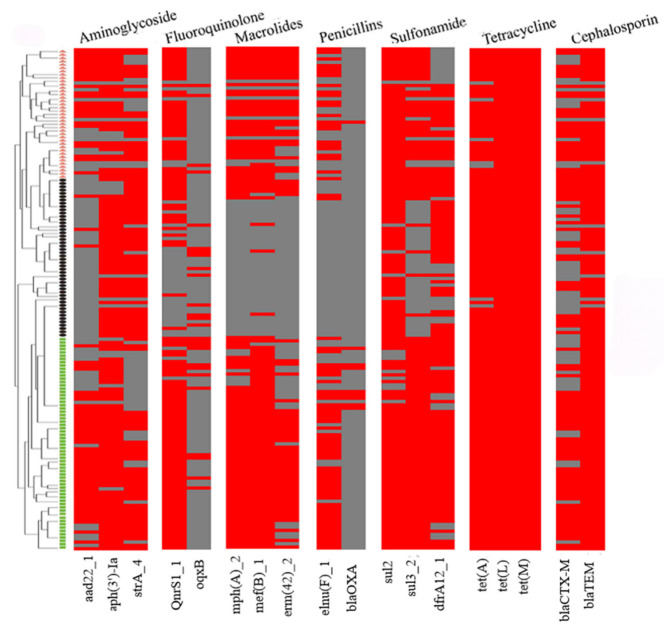
A heatmap showing the distribution of antimicrobial resistance genes among the three *Pasteurellaeae* species. *Pasteurella multocida* strains are indicated using small green boxes; *Glaesserella parasuis* strains are indicated using small black pies; *Actinobacillus pleuropneumoniae* strains are indicated using orange small triangles. Strips in red refer to the presence of an antimicrobial resistance gene while those in gray refer to the absence of an antimicrobial resistance gene.

**Figure 3 microorganisms-13-00938-f003:**
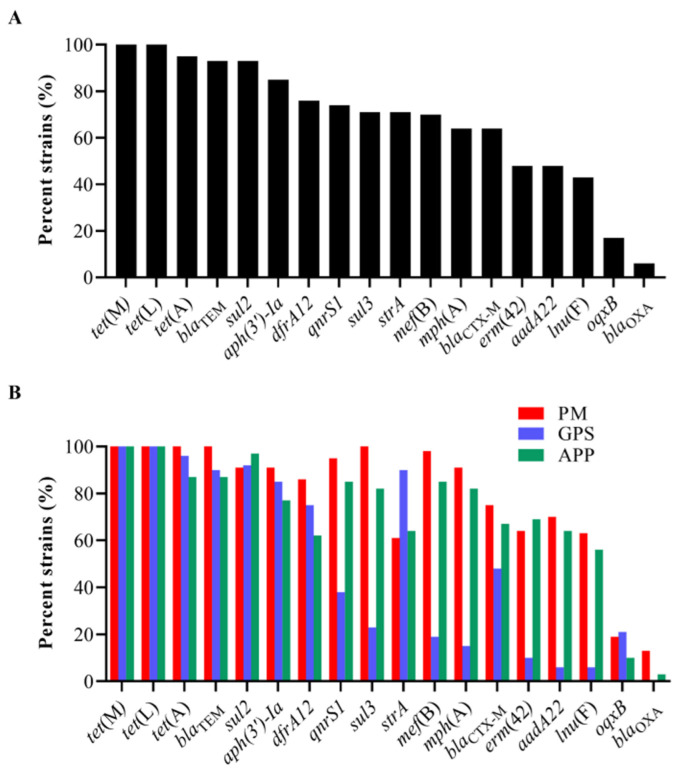
Distribution of antimicrobial resistance genes (ARGs) among the three *Pasteurellaeae* species. (**A**) A column chart showing the detection rates of different ARGs among the 151 *Pasteurellaeae* strains. (**B**) A column chart showing the detection rates of different ARGs among *Pasteurella multocida*, *Glaesserella parasuis*, and *Actinobacillus pleuropneumoniae*.

**Figure 4 microorganisms-13-00938-f004:**
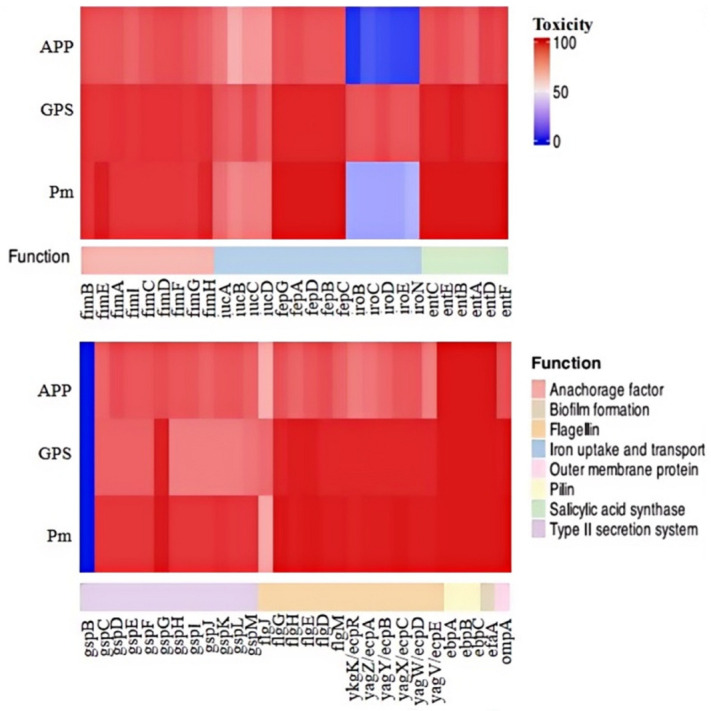
A heatmap showing the distribution of virulence factor genes among the three *Pasteurellaeae* species. PM, *Pasteurella multocida*; GPS, *Glaesserella parasuis*; APP, *Actinobacillus pleuropneumoniae*.

**Table 1 microorganisms-13-00938-t001:** Isolation of the three *Pasteurellaeae* species from pigs in China between 2021 and 2023.

Regions	PM ^1^			GPS			APP		
	2021	2022	2023	2021	2022	2023	2021	2022	2023
Xinjiang	6.67% (2/30)	3.13% (1/32)	0 (0/18)	10.00% (3/30)	0 (0/32)	0 (0/18)	3.33% (1/30)	3.13% (1/32)	0 (0/18)
Sichuan	0.75% (1/134)	0.67% (1/149)	3.49% (3/86)	0.75% (1/134)	0 (0/149)	3.49% (3/86)	1.50% (2/134)	0 (0/149)	2.32% (2/86)
Hunan	0 (0/216)	0.45% (1/221)	0.81% (2/246)	0.46% (1/216)	0.45% (1/221)	0.41% (1/246)	0 (0/216)	0 (0/221)	0.41% (1/246)
Guangxi	1.67% (2/120)	0.87% (1/115)	0.83% (1/120)	0.83% (1/120)	1.74% (2/115)	0.83% (1/120)	0.83% (1/120)	0.87% (1/115)	0 (0/120)
Shandong	2.63% (5/190)	0 (0/171)	1.85% (2/108)	1.05% (2/190)	1.17% (2/171)	1.85% (2/108)	0.53% (1/190)	0.58% (1/171)	0.93% (1/108)
Hebei	9.26% (5/54)	0 (0/56)	12.50% (3/24)	5.56% (3/54)	0 (0/56)	8.33% (2/24)	1.85% (1/54)	3.57% (2/56)	4.17% (1/24)
Gansu	6.82% (3/44)	0 (0/54)	4.65% (4/86)	9.09% (4/44)	0 (0/54)	1.62% (1/86)	2.27% (1/44)	3.70% (2/54)	1.16% (1/86)
Henan	2.22% (1/45)	4.44% (2/45)	3.13% (1/32)	4.44% (2/45)	0 (0/45)	3.13% (1/32)	0 (0/45)	4.44% (2/45)	0 (0/32)
Hubei	0.66% (2/302)	0 (0/330)	3.33% (6/180)	1.32% (4/302)	0.30% (1/330)	0 (0/180)	0.99% (3/302)	0.30% (1/330)	0.56% (1/180)
Shanxi	8.16% (4/49)	0 (0/57)	4.04% (4/99)	0 (0/49)	7.02% (4/57)	1.01% (1/99)	4.08% (2/49)	5.26% (3/57)	1.01% (1/99)
Guangdong	0.48% (1/207)	0.90% (2/221)	0 (0/173)	0.97% (2/207)	0 (0/221)	0 (0/173)	0.48% (1/207)	0.45% (1/221)	0 (0/173)
Yunan	2.00% (1/50)	5.13% (2/39)	1.15% (1/87)	0 (0/50)	2.56% (1/39)	2.30% (2/87)	2.00% (1/50)	5.13% (2/39)	1.15% (1/87)
Total	1.87% (27/1441)	0.67% (10/1490)	2.14% (27/1259)	1.60% (23/1441)	0.74% (11/1490)	1.11% (14/1259)	0.97% (14/1441)	1.07% (16/1490)	0.71% (9/1259)

^1^ PM, Pasteurella multocida; GPS, Glaesserella parasuis; APP, Actinobacillus pleuropneumoniae.

**Table 2 microorganisms-13-00938-t002:** Minimal inhibitory concentration (MIC) values of the tested antibiotics against the three *Pasteurellaeae* species from pigs in China between 2021 and 2023.

Species ^1^	Agents ^2^	No. of Strains with MIC of (μg/mL)												Break Points (μg/mL) ^3^	MIC_50_	MIC_90_	% Resistant
		0.25	0.5	1	2	4	8	16	32	64	128	256	512				
PM	SXT											3	61	NA	512	512	NA
	CEF			18	2	5	0	0	39					NA	32	32	NA
	AMP	3	1	0	0	0	60							2	8	8	93.75%
	TET	0	2	34	0	0	28							2	1	8	43.75%
	TMC					3	4	16	0	0	41			32	128	128	64.06%
	ENR	24	18	0	0	1	21							1	0.5	8	34.38%
	GEN			0	15	26	0	1	22					NA	4	32	NA
GPS	SXT											15	33	NA	512	512	NA
	CEF			10	6	17	5	0	10					NA	4	32	NA
	AMP	0	2	2	0	0	44							NA	8	8	NA
	TET	0	0	9	0	1	38							NA	8	8	NA
	TMC					5	1	23	6	2	11			NA	16	128	NA
	ENR	12	25	0	0	4	7							NA	0.5	8	NA
	GEN			0	0	16	2	0	30					NA	32	32	NA
APP	SXT											9	30	NA	512	512	NA
	CEF			14	12	5	0	2	6					NA	2	32	NA
	AMP	9	2	0	0	1	27							2	8	8	69.23%
	TET	0	2	13	0	0	24							2	8	8	61.54%
	TMC					0	0	16	0	2	21			32	128	128	58.97%
	ENR	24	11	0	0	2	2							1	0.25	8	10.26%
	GEN			0	0	20	0	2	17					NA	4	32	NA

^1^ Bacterial species: PM, *Pasteurella multocida*; GPS, *Glaesserella parasuis*; APP, *Actinobacillus pleuropneumoniae.*
^2^ Antimicrobial agents: AMP: ampicillin, CEF: ceftiofur, ENR: enrofloxacin, GEN: gentamicin, TET: tetracycline, TMC: tilmicosin, SXT: sulfamethoxazole-trimethoprim. ^3^ Breakpoints, veterinary-specific breakpoints applicable to porcine *P. multocida* and *A. pleuropneumoniae* from CLSI VET01S ED7:2024; NA, breakpoint not available.

## Data Availability

Sequence data were deposited in NCBI under BioProjects PRJNA1082799 (*P. multocida*), PRJNA1080278 (*G. parasuis*), and PRJNA1079884 (*A. pleuropneumoniae*).

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
