# Peer review of "Isolation, Antimicrobial Susceptibility, and Genotypes of Three Pasteurellaeae Species Prevalent on Pig Farms in China Between 2021 and 2023"

_microorganisms, 2025, doi:10.3390/microorganisms13040938_

Round 1

Reviewer 1 Report

Comments and Suggestions for Authors

The manuscript with the topic "Isolation, antimicrobial susceptibility, and genotypes of three Pasteurellaeae species prevalent on pig farms in China between 2021 and 2023" (Manuscript ID microorganisms-3552980) is written in a high professional and scientific language.

I would like to ask and suggest some things to be improved in the text:

  1. In RESULTS
    • In my opinion, it was appropriate to add a section with the references for the 18 genes from sections “3. Distribution of antimicrobial resistance genes and their associations with the resistant phenotypes” and “3.4. Distribution of genes associated with bacterial fitness and virulence” in the INTRODUCTION, because You only show them to the reader in line 188;
    • In section “3. Distribution of antimicrobial resistance genes and their associations with the resistant phenotypes” please, check on line 188 is it correct the writing of blaCTX-M, or it must be blaCTX-M-15?;
    • After line 200 on page 6, the numeration of the pages was mistaken until the end of the manuscript;
    • On line 206 aadA22 to be in Italic;
    • Because it would be very long the text it was good to say at least the references from where You took the sequences of the genes/primers or to make a legend with all of them behind the text. As it was presented like this the information in the manuscript related to the genes and primers is not useful for the scientists.
  2. There was no divided CONCLUSION.
  3. REFERENCES: 35% of the references are for a period of the last 5 years. It is mandatory to increase their number.

Self-Citations were 6.5% of all references.

Author Response

Overall comments: The manuscript with the topic "Isolation, antimicrobial susceptibility, and genotypes of three Pasteurellaeae species prevalent on pig farms in China between 2021 and 2023" (Manuscript ID microorganisms-3552980) is written in a high professional and scientific language.

Reply: Thank you very much for your comments and recognition. We appreciate your time and work and we have also revised our manuscript following your valuable suggestions. Below please find our responses point-by-point:

I would like to ask and suggest some things to be improved in the text:

  1. In RESULTS
  • In my opinion, it was appropriate to add a section with the references for the 18 genes from sections “ Distribution of antimicrobial resistance genes and their associations with the resistant phenotypes” and “3.4. Distribution of genes associated with bacterial fitness and virulence” in the INTRODUCTION, because You only show them to the reader in line 188;

Reply: Thank you for the comments. We have added the required information along with the proper citations in the Introduction section. Please see lines 66-76, 91-95.

  • In section “ Distribution of antimicrobial resistance genes and their associations with the resistant phenotypes” please, check on line 188 is it correct the writing of blaCTX-M, or it must be blaCTX-M-15?;

Reply: Thank you for pointing this out for us. We have double checked the bioinformatical results and confirmed it to be blaCTX-M.

  • After line 200 on page 6, the numeration of the pages was mistaken until the end of the manuscript;

Reply: Sorry about this and we have revised the page numbers and have made them correctly.

  • On line 206 aadA22to be in Italic;

Reply: Thank you for pointing this out for us. We have revised it in our whole manuscript. See line 246.

  • Because it would be very long the text it was good to say at least the references from where You took the sequences of the genes/primers or to make a legend with all of them behind the text. As it was presented like this the information in the manuscript related to the genes and primers is not useful for the scientists.

Reply: We just followed the primers and protocols published by previous articles and we have added them in the manuscript. Please check line 121. Thank you.

  1. There was no divided CONCLUSION.

Reply: A separate conclusion section has been added. Please check lines 375-386. Thank you.

  1. REFERENCES: 35% of the references are for a period of the last 5 years. It is mandatory to increase their number.

Reply: We have updated our reference list. Please check it. Thank you for your suggestion.

Reviewer 2 Report

Comments and Suggestions for Authors

This study investigates the prevalence, antimicrobial susceptibility, and genetic characteristics of Pasteurella multocida, Glaesserella parasuis, and Actinobacillus pleuropneumoniae on pig farms in China between 2021 and 2023. A total of 151 bacterial strains were isolated, and antimicrobial susceptibility testing, along with whole-genome sequencing, was performed. The study identifies dominant serotypes, high antimicrobial resistance rates, and multiple resistance genes. These findings provide valuable insights into the epidemiology of these bacterial pathogens, which have significant implications for swine health and antibiotic stewardship.

SIMPLE SUMMARY

  • The manuscript lacks a Simple Summary, which is an essential component for broad readership, especially for non-specialists or practitioners in the field. A Simple Summary should provide a concise, accessible overview of the study, avoiding technical jargon while maintaining scientific accuracy.

ABSTRACT

  • The abstract lacks a clear hypothesis or research question. Instead of being purely descriptive, it should specify the study's objective in a testable manner.

INTRODUCTION

  • The introduction does not clearly state the knowledge gap this study aims to fill. While it highlights the importance of AMR surveillance, it does not specify what is unknown or uncertain about these pathogens in China. Define the knowledge gap more clearly.

MATERIALS AND METHODS

  • Statistical methods are not mentioned. Were resistance rates compared using statistical tests? If not, this should be added.

RESULTS

  • The results are presented descriptively but lack statistical comparisons. Were there significant differences in resistance between species or years?
  • The association between genotype (ARGs) and phenotype (resistance patterns) is mentioned, but no quantitative measure (e.g., correlation coefficients) is provided.

DISCUSSION

  • The study follows standard protocols (e.g., CLSI guidelines), but the methodology for G. parasuis antimicrobial susceptibility testing lacks CLSI-approved breakpoints. It would be beneficial to discuss how the chosen interpretation criteria affect the results.
  • No mention of study limitations. Were there biases in sample collection, methodological constraints, or limitations in sequencing coverage?
  • While the study highlights resistance rates, it lacks a discussion on the clinical implications of these findings for swine disease management. How should veterinarians adapt treatment strategies based on these results?

CONCLUSION

  • The manuscript does not have a clearly defined Conclusion section. While some concluding remarks are present in the final paragraphs of the Discussion, they are not distinctly separated as a Conclusion, which is a standard requirement in scientific papers.

  • The conclusion is too general and does not clearly state the study's contributions. What new insights does this study provide compared to previous research?
  • There is no mention of future research directions. Should surveillance continue in other regions? Should alternative control strategies be investigated?

Author Response

Overall comments: This study investigates the prevalence, antimicrobial susceptibility, and genetic characteristics of Pasteurella multocidaGlaesserella parasuis, and Actinobacillus pleuropneumoniae on pig farms in China between 2021 and 2023. A total of 151 bacterial strains were isolated, and antimicrobial susceptibility testing, along with whole-genome sequencing, was performed. The study identifies dominant serotypes, high antimicrobial resistance rates, and multiple resistance genes. These findings provide valuable insights into the epidemiology of these bacterial pathogens, which have significant implications for swine health and antibiotic stewardship.

Reply: Thank you very much for your comments and recognition. We appreciate your time and work and we have also revised our manuscript following your valuable suggestions. Below please find our responses point-by-point:

SIMPLE SUMMARY

  • The manuscript lacks a Simple Summary, which is an essential component for broad readership, especially for non-specialists or practitioners in the field. A Simple Summary should provide a concise, accessible overview of the study, avoiding technical jargon while maintaining scientific accuracy.

Reply: Thank you very much for your comments. It seems that a Simple Summary is NOT a required part by the journal Microorganisms as I did not find the related contents from the author submission guidelines as well as from the published articles. I am not sure whether this part is necessary and what formats or lengths should I prepare. Anyway, we have added a simple summary less than 120 words as suggested but if the editorial office think it is not necessary it could be deleted directly. Please check lines 16-24.   

ABSTRACT

  • The abstract lacks a clear hypothesis or research question. Instead of being purely descriptive, it should specify the study's objective in a testable manner.

Reply: Thank you very much for your suggestions. We have highlighted the objective of the study in the Abstract section in Blue. Please check lines 27-28.

INTRODUCTION

  • The introduction does not clearly state the knowledge gap this study aims to fill. While it highlights the importance of AMR surveillance, it does not specify what is unknown or uncertain about these pathogens in China. Define the knowledge gap more clearly.

Reply: Thank you very much for your suggestions. We have rephrased our Introduction section to make it more clearer. The knowledge gap has been stated at lines 100-103.

MATERIALS AND METHODS

  • Statistical methods are not mentioned. Were resistance rates compared using statistical tests? If not, this should be added.

Reply: Good suggestions. Please see the statistical analysis section from lines 167-170. Thank you.

RESULTS

  • The results are presented descriptively but lack statistical comparisons. Were there significant differences in resistance between species or years?
  • The association between genotype (ARGs) and phenotype (resistance patterns) is mentioned, but no quantitative measure (e.g., correlation coefficients) is provided.

Reply: Thank you very much for your suggestions. We have added the statistical analysis for the results when applicable. Please see lines 206-208. In addition, we did not observe associations between the presence of ARGs and the resistant phenotypes was observed in this study (lines 251-252). This agrees well with previous studies (e.g., Antibiotics (Basel) 2020, 9(9):614; 46.Microbiol Spectr 2018, 6(3):10).

DISCUSSION

  • The study follows standard protocols (e.g., CLSI guidelines), but the methodology for  parasuisantimicrobial susceptibility testing lacks CLSI-approved breakpoints. It would be beneficial to discuss how the chosen interpretation criteria affect the results.
  • No mention of study limitations. Were there biases in sample collection, methodological constraints, or limitations in sequencing coverage?
  • While the study highlights resistance rates, it lacks a discussion on the clinical implications of these findings for swine disease management. How should veterinarians adapt treatment strategies based on these results?

Reply: Thank you very much for your suggestions. We have rephrased our discussion sections and have added the required information. For the discussion of the lacking CLSI breakpoints for G. parasuis, please see lines 341-346. For the statement of limitations, please see lines 367-374. For the discussion on the clinical implications, please see lines 346-351.   

CONCLUSION

The manuscript does not have a clearly defined Conclusion section. While some concluding remarks are present in the final paragraphs of the Discussion, they are not distinctly separated as a Conclusion, which is a standard requirement in scientific papers.

  • The conclusion is too general and does not clearly state the study's contributions. What new insights does this study provide compared to previous research?
  • There is no mention of future research directions. Should surveillance continue in other regions? Should alternative control strategies be investigated?

Reply: A separate conclusion section has been added. Please check lines 375-386. Thank you.

Round 2

Reviewer 2 Report

Comments and Suggestions for Authors

Thank you very much for your clarification regarding the Simple Summary. You are indeed correct—after revisiting the author guidelines and several published articles in Microorganisms, it appears that including a Simple Summary is not a mandatory requirement for this journal. I also appreciate your decision to include a concise summary of fewer than 120 words as a precautionary measure.

Thank you as well for the additional revisions. The manuscript is now much clearer, well-structured, and substantially improved in both content and presentation.